# Programming bacteria for multiplexed DNA detection

Yu-Yu Cheng[1,4], Zhengyi Chen [1,4], Xinyun Cao[1], Tyler D. Ross[1], Tanya G. Falbel[2], Briana M. Burton [2] & Ophelia S. Venturelli [1,2,3] ✉

DNA is a universal and programmable signal of living organisms. Here we develop cell-based DNA sensors by engineering the naturally competent bacterium *Bacillus subtilis* (*B. subtilis*) to detect specific DNA sequences in the environment. The DNA sensor strains can identify diverse bacterial species including major human pathogens with high specificity. Multiplexed detection of genomic DNA from different species in complex samples can be achieved by coupling the sensing mechanism to orthogonal fluorescent reporters. We also demonstrate that the DNA sensors can detect the presence of species in the complex samples without requiring DNA extraction. The modularity of the living cell-based DNA-sensing mechanism and simple detection procedure could enable programmable DNA sensing for a wide range of applications.

Next-generation engineered bacteria hold tremendous promise for a wide range of applications in human health, environment and agriculture. These bacteria can sense key environmental signals and perform computation on these signals to regulate a response that modulates specific environmental parameters[1]. Developing specific and selective sensors of key environmental signals is a critical feature of next-generation engineered bacteria. For example, bacteria have been engineered to detect physical and chemical signals such as light, ultrasound, and quorum-sensing molecules[2–4]. These signals could be exploited to control the collective growth or gene expression of the bacterial population or mediate interactions between constituent community members[4–6]. In addition, we could exploit their sensing ability to achieve real-time monitoring of natural environments. For example, synthetic genetic circuits have been designed in bacteria to sense signals produced by pathogens and use this information to regulate the production of antimicrobials that inhibit the specific pathogen[7,8]. However, there are limited well-characterized and orthogonal signals that can be exploited to sense different bacterial species in a microbial community[9,10].

DNA provides the blueprint for living organisms and is prevalent in natural environments[11]. Therefore, extracellular DNA (eDNA) could be exploited as a biomarker for identifying different species. Naturally competent bacteria have the capability to take up DNA from the environment and integrate these imported sequences into the genome based on sequence homology requirements. Horizontal gene transfer (HGT) via natural transformation can facilitate nutrient utilization, DNA repair, or the acquisition of new genes[12]. Since homologous recombination of imported DNA requires sequences of sufficient length and homology[13], natural transformation could be exploited to build a selective cell-based DNA sensor.

We constructed a living cell-based DNA sensor by designing a novel synthetic genetic circuit in the naturally competent bacterium *B. subtilis*. This circuit controls *B. subtilis* growth and fluorescence reporter genes in response to specific input DNA sequences. We demonstrate that the cell-based DNA sensor is highly specific to species harboring the target DNA sequence. In addition, we show that cell-based DNA sensors can perform multiplexed DNA detection in complex samples. The cell-based DNA sensors can also detect DNA released from pre-treated target cells. Our detailed characterization of the cell-based DNA sensors in vitro provides a foundation for future DNA-sensing applications.

## Results

### Construction of a living DNA sensor strain
To build the living cell-based DNA sensor, we exploited the natural competence ability of the well-characterized soil bacterium *B. subtilis*[14]. The natural competence ability of *B. subtilis* enables uptake of environmental DNA and integration of specific sequences with

[1]Department of Biochemistry, University of Wisconsin-Madison, Madison, WI, USA. [2]Department of Bacteriology, University of Wisconsin-Madison, Madison, WI, USA. [3]Department of Chemical & Biological Engineering, University of Wisconsin-Madison, Madison, WI, USA. [4]These authors contributed equally: Yu-Yu Cheng, Zhengyi Chen. ✉e-mail: venturelli@wisc.edu

sufficient homology onto the genome via homologous recombination[15]. The efficiency of homologous recombination depends stringently on the sequence percent identity and length[13,16], which can be exploited to build a highly specific DNA sensor.

To detect eDNA sequences in a programmable fashion, we constructed a synthetic genetic circuit in *B. subtilis* that implements a growth selection function based on the presence of target DNA sequence in the environment. The circuit consists of a xylose-inducible master regulator of competence *comK*[17] and an isopropyl β-d-1 thio-galactopyranoside (IPTG)-inducible toxin-antitoxin system *txpA-ratA*[18] and green fluorescent protein (GFP) regulated by the repressor LacI (Fig. 1a; Supplementary Fig. 1). The flanking regions (upstream and downstream) of *txpA-ratA* and *lacI* consist of the target sequences and are referred to as landing pads for homologous recombination.

In the presence of xylose, ComK activates competence genes for DNA uptake and homologous recombination (Fig. 1a). Bistability and stochastic processes in the regulation of natural competence can yield a sub-population that can be transformed with extracellular DNA[19]. This naturally competent sub-population forms competence pili and binds to double stranded DNA outside the cell. The DNA is degraded into single-stranded DNA (ssDNA) outside the cell membrane, and transported into the cell[12]. Inside the cell, RecA binds the ssDNA and searches the *B. subtilis* genome for a region with sufficient homology. If the target DNA sequence is present, homologous recombination removes the toxin-antitoxin *txpA-ratA* and repressor

*lacI*. To select for the transformed subpopulation, the introduction of IPTG induces the expression of the toxin TxpA, which in turn inhibits the growth of the non-transformed subpopulation by blocking cell wall synthesis[18]. The transformed sub-population can grow and express GFP (Fig. 1a).

We constructed a sensor for *Escherichia coli* (*E. coli*) (EC sensor) by introducing the *xdhABC* operon onto the *B. subtilis* genome (landing pad region), which encodes genes for purine catabolism[20]. We analyzed the degree of conservation of the target sequence by evaluating sequences of different *E. coli* strains using the BLAST search in the NCBI Nucleotide Collection Database. This sequence is highly conserved such that 99% of 5000 *E. coli* genomes in the NCBI database contain this sequence with >95% coverage (the degree of alignment of the query sequence with a reference sequence) and >95% identity similarity (the percentage of bases that are identical to the target sequence within the aligned region). Therefore, the *xdhABC* operon is a representative sequence that can detect a wide range of *E. coli* strains.

To characterize the homology length needed for robust DNA sensing, we varied the homology length of the *xdhABC* operon in each landing pad (0.5–2.5 kb). We performed time-series measurements of transformation efficiency with 100 ng/mL *E. coli* genomic DNA (gDNA). The transformation efficiency is defined as the ratio of the number of transformed *B. subtilis* to the total number of *B. subtilis* based on colony forming units (CFU). Transformation efficiency plateaued at ~10 h and the colonies expressed GFP (Fig. 1b; Supplementary Fig. 2a).

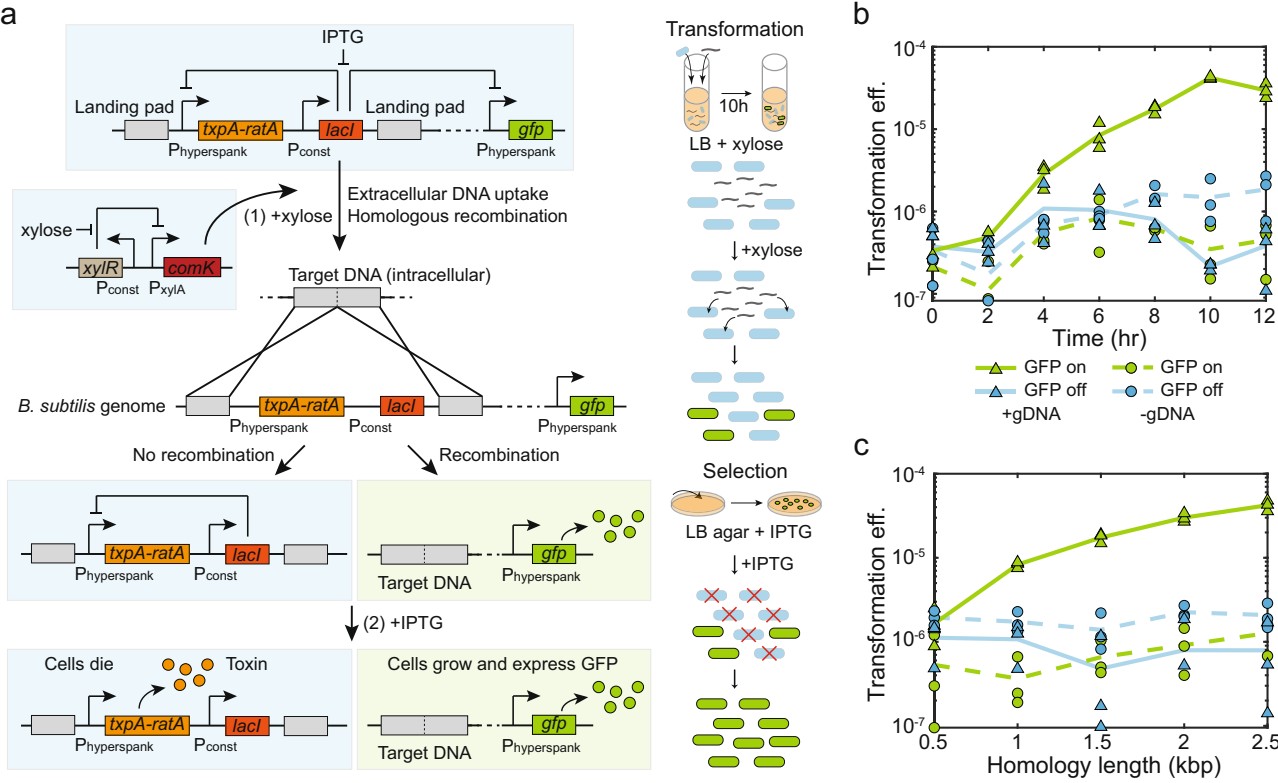

**Fig. 1 | Construction of a living cell-based DNA sensor. a** Schematic of a synthetic genetic circuit constructed in *B. subtilis* that allows the recognition of specific extracellular DNA sequences. The xylose-inducible master regulator of competence *comK* induces *B. subtilis* natural competence. The repressor LacI regulates the expression of the toxin-antitoxin system *txpA-ratA* and fluorescent protein GFP. The flanking regions of the toxin and LacI repressor consist of the target sequences. In the presence of target DNA, homologous recombination will remove the toxin and the repressor, allowing a sub-population of transformed cells to express GFP. To select the transformed cells, the addition of IPTG addition inhibits the growth of the non-transformed sub-population via the induction of the toxin, allowing the transformed sub-population to grow and express GFP. **b** Line plot of

transformation efficiency of *E. coli* sensor (2.5 kb landing pad) versus transformation time in the presence (solid lines) or absence (dashed lines) of 100 ng/mL purified *E. coli* gDNA. Transformed cells were fluorescent and the number increased as a function of time. The observed transformants in the absence of gDNA were due to escape mutations. **c** Line plot of transformation efficiency of the *E. coli* sensor versus the landing pad length in the presence (solid lines) or absence (dashed lines) of 100 ng/mL purified *E. coli* gDNA. The transformation efficiency was above the background when the landing pad was greater than or equal to 1 kb and increased as a function of homology length. Data points represent three biological replicates and lines are the average of replicates. Source data are provided as a Source Data file.

In addition, transformation efficiency at 10 h increased with homology length in the landing pads (Fig. 1c). A homology length of 1 kb or greater was required to robustly sense the target sequence over the background frequency of escape mutants ($10^{-7}$–$10^{-6}$ frequency) that displayed heterogenous GFP expression in the absence of gDNA (Supplementary Fig. 2b). To achieve high performance of the DNA sensor (>$10^2$ increase in transformation efficiency above background), we used a landing pad homology length of 2.5 kb (transformation efficiency of $10^{-5}$–$10^{-4}$). In the transformed sub-population, homologous recombination occurred at the expected location with the elimination of *txpA-ratA* and *lacI* based on sequencing (Supplementary Fig. 2c, d). The moderate number of observed escape mutants that displayed growth in the absence of gDNA had mutations in *txpA* or *lacI*, which reduced the growth inhibitory activity of TxpA (Supplementary Fig. 2e, f). In sum, the synthetic genetic circuit enabled *B. subtilis* to sense a specific *E. coli* DNA sequence present in the environment.

### Building living DNA sensors to sense human pathogens

Exploiting the modularity of the DNA sensing circuit, we replaced the landing pad region with specific sequences targeting different bacterial strains (Supplementary Fig. 1c). To this end, we constructed DNA sensors to detect sequences harbored in human gut pathogens *Salmonella typhimurium*[21] (*S. typhimurium*), *Clostridioides difficile*[22] (*C. difficile*), and the skin and respiratory tract pathogen *Staphylococcus aureus*[23] (*S. aureus*). We selected two 2.5 kb sequences in the pathogenicity island *sipBCDA* in *S. typhimurium* (ST sensor), the heme biosynthesis pathway *hemEH* in *S. aureus* (SA sensor), and the phenylalanyl-tRNA synthetase *pheST* in *C. difficile* (CD sensor)[24–26].

The selected set of target DNA sequences are highly conserved within a given species (ST sensor: 94%, SA sensor: 96%, and CD sensor: 96% all with >95% coverage and >95% identity similarity). In addition, some of the sequences are linked to virulence activities of the pathogen or encode enzymes that are critical for fitness[24–26]. To further explore the conservation of the target sequences across different strains, we performed nucleotide BLAST using the NCBI Database to quantify the homology coverage and sequence similarity across species. The pathogenicity island *sipBCDA* in *S. typhimurium* was found only in genomes of *Salmonella enterica* and infrequently observed in other species (Supplementary Fig. 3a). Homologs in other species have low coverage and identity similarity, suggesting that the pathogenicity island could be a good target sequence for this species (Supplementary Fig. 3a). The heme biosynthesis pathway *hemEH* in *S. aureus* and phenylalanyl-tRNA synthetase *pheST* in *C. difficile* are conserved in some closely related strains with varying degrees of similarity and coverage (Fig. 2i, k). The *E. coli xdhABC* purine catabolism operon is found in other closely related bacteria such as *Shigella* with high coverage and identity similarity (Supplementary Fig. 3b). Although the target sequences for building the different DNA sensor strains varied in the degree of specificity based on bioinformatic analyses, a detailed characterization of DNA sensor performance could nevertheless guide the design of optimized cell-based DNA sensors for future applications.

The four sensors robustly detected the presence of 100 ng/mL target gDNA over background (no gDNA) based on transformation efficiency (Fig. 2a). We evaluated the sensitivity of each DNA sensor strain by performing time-series fluorescence measurements in selective liquid media after being transformed with a wide range of gDNA concentrations (0–1500 ng/mL gDNA) (Fig. 2b, c; Supplementary Fig. 4). We evaluated the time required for each culture to display a fluorescence level higher than a threshold (i.e. detection time) as a metric for DNA sensor performance (Fig. 2c; Supplementary Fig. 4). The sensitivity of the sensor strain was defined as the lowest gDNA concentration that yielded a statistically significant difference in the detection time in the presence versus absence of gDNA (Fig. 2d–g). The EC, SA, and CD DNA sensor strains displayed high sensitivity of 1–16 ng/mL ($10^5$–$10^6$ chromosome copy number per mL), whereas the ST sensor displayed a lower sensitivity (62.5 ng/mL, $10^7$ chromosome copy number per mL).

The relationship between the log transformed gDNA concentration and detection time is linear due to the exponential growth of the fluorescent *B. subtilis* sub-population in the selective liquid medium (Supplementary Fig. 5). Therefore, to assess the range of gDNA concentrations that was accurately sensed, we inferred the parameters of a linear function fit to the log transformed gDNA concentration versus detection time (Fig. 2d–g). The inferred slope of the linear function is determined by the cell doubling time (~0.5 h) and intercept is determined by the background mutation frequency (Fig. 2d–g). The observed goodness of fit of the data to a linear function indicates that DNA sensing is quantitative across a wide range of gDNA concentrations.

To characterize the specificity of each DNA sensor strain to the target sequence, we performed time-series fluorescence measurements in liquid culture in response to the four individual species' gDNA. The fluorescence signal was observed at a substantially earlier time (6.1–7.1 h) in the presence of the corresponding species' gDNA than in the presence of a non-target species' gDNA or in the absence of DNA (9.1–10.7 h) (Fig. 2h; Supplementary Fig. 6). To further examine the specificity, we evaluated the ability of the DNA sensors to distinguish between closely related species with similar target sequences. To this end, we measured the transformation efficiency of the SA sensor in the presence of gDNA (100 ng/mL) derived from *S. epidermidis*. *S. epidermidis* is a closely related human skin commensal bacterium that harbors a *hemEH* homolog (89% coverage and 77% identity similarity) (Fig. 2i). To simplify the procedure, we quantified the number of transformed colonies as opposed to transformation efficiency since the number of total *B. subtilis* colonies was not substantially affected by the presence or absence of gDNA (Fig. 2a; Supplementary Fig. 7). The number of transformed SA sensor colonies in the presence of *S. epidermidis* gDNA (SE) was similar to the absence of DNA and substantially lower than in the presence of *S. aureus* gDNA (SA) (Fig. 2j). Hence, the SA sensor can distinguish the target pathogen *S. aureus* DNA from a closely related commensal *S. epidermidis* DNA. Similarly, the CD sensor can distinguish the target pathogen *C. difficile* DNA from a closely related commensal species *C. hiranonis* DNA harboring a *pheST* homolog (87% coverage and 75% identity similarity) (Fig. 2k, l). These data demonstrate that the cell-based DNA sensors were highly specific to the target sequence.

### Multiplexed detection of pathogen DNA in complex samples

Since certain future applications may require sensing of more than one organism, we tested the ability of the DNA sensors to detect more than one species within mixed DNA samples. To this end, we constructed individual sensors with orthogonal fluorescent reporters to achieve multiplexed DNA detection. Exploiting the modularity of the circuit, we constructed an RFP-labeled ST sensor (ST-R) and a BFP-labeled SA sensor (SA-B), in addition to the GFP-labeled EC sensor (EC-G) (Supplementary Fig. 1c). We introduced gDNA (200 ng/mL) extracted from each of the three target strains into a mixture containing EC-G, ST-R, and SA-B sensors and determined the number of fluorescent colonies for each reporter (Fig. 3a). The sensors accurately reported the presence/absence of species' gDNA reliably for all combinations (Fig. 3b; Supplementary Fig. 8). Therefore, a mixture of DNA sensor strains enabled multiplexed detection of gDNA with high specificity.

To investigate if multiplexed detection can be achieved for samples derived from a complex microbial community, we constructed a four-member human gut community composed of diverse commensal bacteria from three major phyla in human gut—*Anaerostipes caccae* (AC, Firmicutes), *Bacteroides thetaiotaomicron* (BT, Bacteroidetes), *Bifidobacterium longum* (BL, Actinobacteria), and *Clostridium asparagiforme* (CG, Firmicutes) (Fig. 3c). This community also contained the

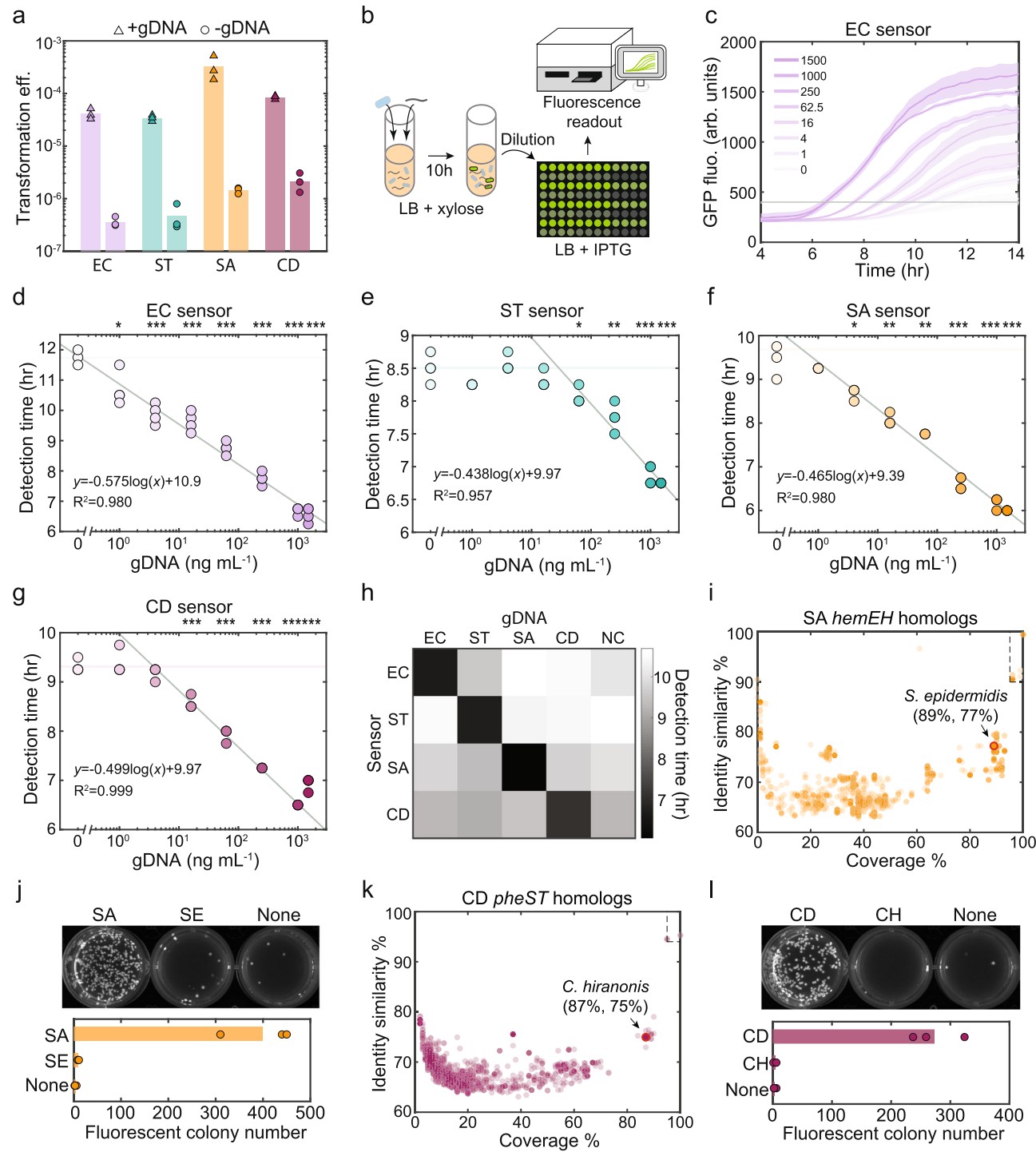

target pathogens *S. typhimurium* (ST) and *S. aureus* (SA). We tested whether the DNA sensors could accurately report the relative abundance of the two pathogens during community assembly. The 6-member community was inoculated in equal initial species proportions based on absorbance at 600 nm (OD600, Day 0) and cultured anaerobically for 24 h (Day 1). An aliquot of the community was transferred to fresh media and community composition was characterized following an additional 24 h (Day 2). Based on 16 S rRNA gene sequencing, the abundance of *S. typhimurium* was similar as a function of time whereas the abundance of *S. aureus* decreased over time (Fig. 3c).

We characterized the ability of the DNA sensors to accurately track the temporal trends in species abundance by introducing

purified community gDNA collected at different times into a mixed culture of the ST-R and SA-G sensors (Fig. 3d). Due to the low abundance of target species in the sample, a higher amount of DNA (1 μg/mL) was used for transformation. Consistent with the trends based on 16 S rRNA gene sequencing, the number of green fluorescent colonies of the SA-G sensor displayed a decreasing trend, whereas the number of red fluorescent colonies of ST-R sensor were similar over time (Fig. 3d–f). The SA sensor displayed better performance in mirroring the trend based on 16S rRNA gene sequencing than the ST sensor, consistent with its higher sensitivity than other sensors (Figs. 2a, 3e, f). For the community lacking *S. typhimurium* and *S. aureus*, a much smaller number of background mutant colonies was detected than in the presence of the 6-member community

**Fig. 2 | Cell-based DNA sensors can detect major human pathogens.**
**a** Transformation efficiency of cell-based DNA sensors that can detect *E. coli* (EC sensor), *S. typhimurium* (ST sensor), *S. aureus* (SA sensor), or *C. difficile* (CD sensor) in the presence of 100 ng/mL gDNA (triangles) or absence of DNA (circles). Bar represents the average of three biological replicates. **b** Schematic of experimental procedure for time-series fluorescence measurements of transformed sensors. **c** Time-series measurements of GFP expression of EC sensor in liquid medium after being transformed with varying *E. coli* gDNA concentrations (0 to 1500 ng/mL). A threshold of GFP fluorescence 400 arbitrary units (a.u.) was used to determine the detection time for each gDNA concentration. Lines denote the average of four technical replicates and shaded regions represent one standard deviation from the average. The gDNA concentration versus the detection time for the (**d**) EC sensor, (**e**) ST sensor, (**f**) SA sensor and (**g**) CD sensor. Horizontal line represents the detection time of background escape mutants in the absence of gDNA. An unpaired *t*-test was performed to determine if the detection time at a specific DNA concentration was significantly different from the detection time of escape mutants, and *, **, and *** denote *p*-values < 0.05, 0.01 and 0.001, respectively. A linear function was fit to the gDNA versus detection time data points that were significantly different. The slope of the fitted line was determined by the cell growth of transformed *B. subtilis* and the intercept was determined by the background mutation rate. The coefficient of determination $R^2$ indicates the goodness of the fit. **h** Heatmap of detection time of each DNA sensor after being transformed with 100 ng/mL gDNA extracted from a given target species or no gDNA as a negative

control (NC). The detection time of target DNA was lower than non-target DNA or no DNA. Data are the average of four technical replicates. **i** Scatter plot of percent coverage versus identity similarity based on a nucleotide BLAST search of 5000 bp *S. aureus hemEH* in the NCBI database. Each circle represents a homolog with specific coverage and identity similarity found in a given species excluding *S. aureus*. A closely related human commensal *Staphylococcus epidermidis* (*S. epidermidis*) with high coverage and identity similarity was selected to evaluate the specificity of the SA sensor. The region within dashed lines are *Staphylococcus* genomes predicted to be recognized by the SA sensor based on the high conservation of the target sequence. **j** Fluorescent colony numbers of SA sensor on selective agar plates after being transformed with *S. aureus* gDNA (100 ng/mL), *S. epidermidis* gDNA (100 ng/mL), or no gDNA. Bar represents the average of three technical replicates. **k** Scatter plot of percent coverage versus identity similarity based on a nucleotide BLAST search of 5000 bp *C. difficile pheST* in the NCBI database. Each circle represents a homolog with specific coverage and identity similarity found in a given species excluding *C. difficile*. A closely related human commensal *Clostridium hiranonis* (*C. hiranonis*) with high coverage and identity similarity was selected to evaluate the specificity of the CD sensor. The region within dashed lines are *Clostridium* species predicted to be recognized by the CD sensor. **l** Fluorescent colony numbers of the CD sensor on selective agar plates after being transformed with *C. difficile* gDNA (100 ng/mL), *C. hiranonis* gDNA (100 ng/mL), or no gDNA. Bar represents the average of three technical replicates. Source data are provided as a Source Data file.

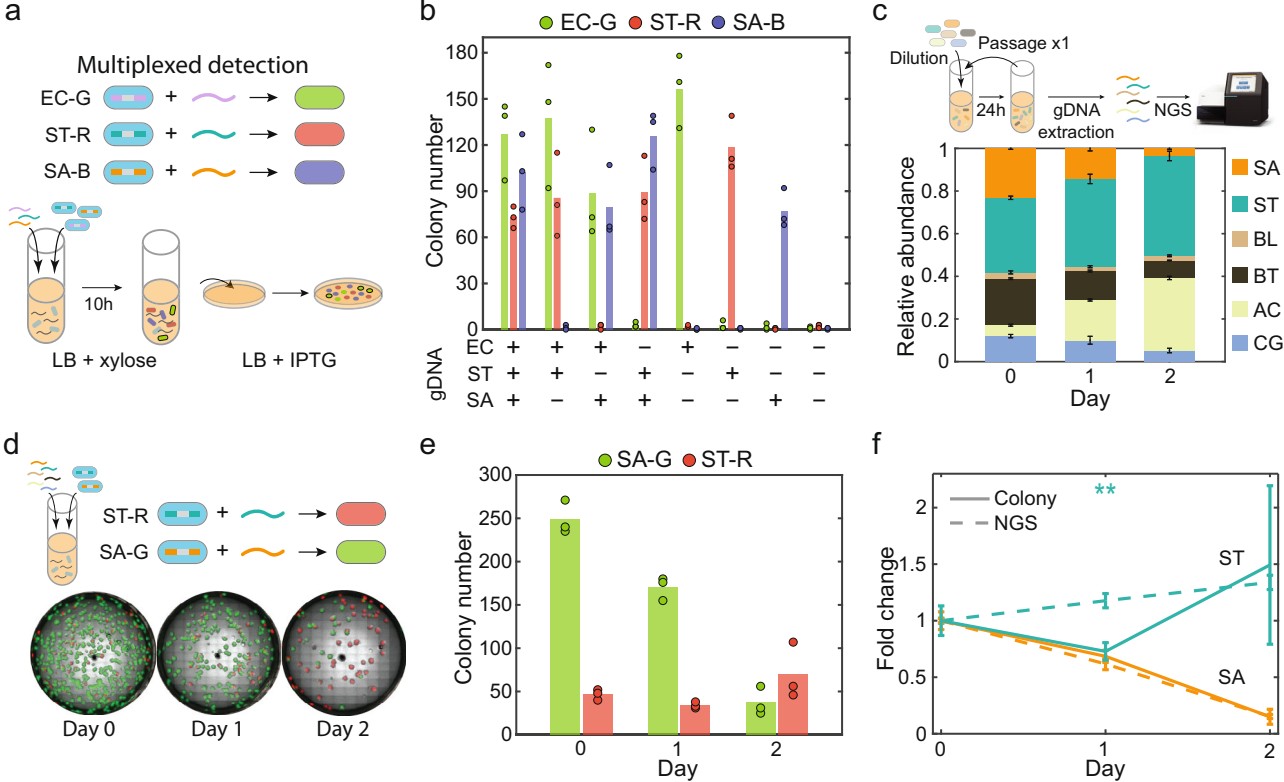

**Fig. 3 | Cell-based DNA sensors can perform multiplexed detection in complex samples. a** Schematic of experimental procedure for multiplexed DNA detection. *E. coli*, *S. typhimurium*, and *S. aureus* DNA sensors were labeled with GFP (EC-G), RFP (ST-R) and BFP (SA-B), respectively. The gDNA extracted from three strains was introduced into a mixed culture of sensors for transformation and selection. The appearance of colonies expressing GFP, RFP or BFP can indicate the presence of target DNA. **b** Bar plot of the numbers of GFP, RFP, or BFP positive colonies on selective agar plates after a mixed culture of EC-G, ST-R, and SA-G was transformed with combinations of different species' gDNA (200 ng/mL each). Sensors precisely detected the presence/absence of target species' gDNA. Bar represents the average of three technical replicates. **c** Relative abundance of six species in a synthetic microbial community composed of *S. aureus* (SA), *S. typhimurium* (ST), *B. longum* (BL), *B. thetaiotaomicron* (BT), *A. caccae* (AC), and *C. asparagiforme* (CG) after

2 days of co-culture in anaerobic conditions. Species relative abundance was determined by 16S rRNA gene next-generation sequencing. Bar height represents the average of three technical replicates of relative abundance and error bars denote one standard deviation. **d** Fluorescence images of transformed SA-G sensor and ST-R sensor colonies on selective agar plates after being transformed with community gDNA collected on different days (1000 ng/mL each). **e** Bar plot of the numbers of GFP or RFP positive colonies on selective agar plates. Bar height represents the average of three technical replicates. **f** Fold change in the abundance of *S. aureus* and *S. typhimurium* over time based on 16S rRNA gene sequencing (dashed line) or cell-based detection (solid line). The fold change of *S. typhimurium* on day 1 was significantly different based on a unpaired *t*-test (*p*-value = 0.0015). Lines represent the average of three technical replicates and error bars denote one standard deviation. Source data are provided as a Source Data file.

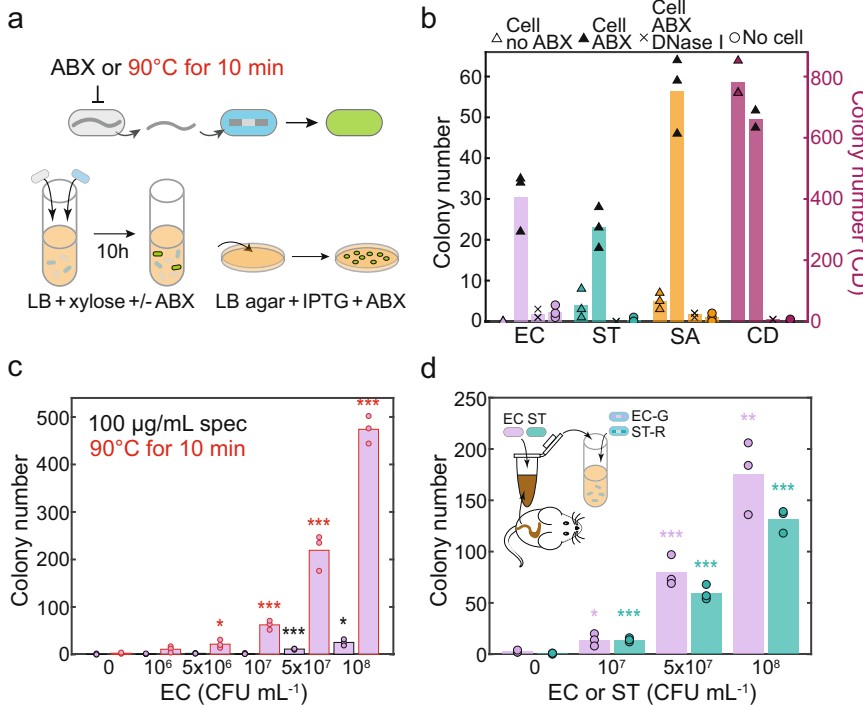

**Fig. 4 | Cell-based DNA sensors can detect species without DNA extraction.**
**a** Schematic of experimental design for DNA detection in the co-culture. Addition of selective antibiotics (ABX) in the co-culture or pre-treated target strain by heat killing can enhance gDNA release and detection by the cell-based DNA sensor.
**b** Bar plot of colony numbers of transformed EC, ST, SA, and CD sensors co-cultured with target strain only (open triangle), target strain and spectinomycin (100 μg/mL) (solid triangle), target strain, spectinomycin, and 1 unit/mL DNase I (x), or spectinomycin only (open circle). Spectinomycin enhanced the detection of *E. coli*, *S. typhimurium*, and *S. aureus* in the co-cultures but was not required for the detection of *C. difficile*. Addition of DNase I in the co-culture substantially reduced the number of transformed sensor colonies, indicating that the detection was via natural transformation. **c** Bar plot of colony numbers of EC sensor co-cultured with varying amounts of spectinomycin-treated or heat-treated *E. coli*. Heat treatment

can significantly increase the transformation efficiency and achieve a detection limit of 5∗10⁶ cells per mL. **d** Bar plot of colony numbers of EC-G and ST-R sensors cultured with 10 mg mouse cecal contents containing varying amounts of spike-in *E. coli* and *S. typhimurium*. Mouse cecal contents were derived from germ-free mice orally gavaged with a synthetic microbial community. The sample (mouse ceca and spike-in target strains) was heat-treated and transferred to the mixed culture of EC-G and ST-R sensors. The detection limit for *E. coli* and *S. typhimurium* was 10⁷ CFU per mL or 10⁹ CFU per gram. An unpaired *t*-test was performed to determine if the colony number in the presence of a specific target strain was significantly different than the negative control (no target cells), and *, **, and *** denote *p*-values < 0.05, 0.01, and 0.001, respectively. Bar height represents the average of three technical replicates. Source data are provided as a Source Data file.

gDNA (Supplementary Fig. 9). This implies that the ST and SA sensors were specific to the target species gDNA and did not recognize the other constituent community member gDNA. In sum, our results show that accurate multiplexed DNA detection can be achieved in DNA samples derived from multi-species microbial communities.

**Detection of target species without DNA extraction**
Specific bacterial species have been shown to release eDNA in response to environmental stimuli[11], suggesting that the DNA sensor could detect species without requiring prior gDNA purification. To test this possibility, we co-cultured individual DNA sensor strains with the corresponding target species with an initial OD600 0.1 of the target strain ($1.22 \times 10^8$ CFU/mL, $1.07 \times 10^8$ CFU/mL, $3.2 \times 10^8$ CFU/mL, and $1.1 \times 10^7$ CFU/mL for *E. coli*, *S. typhimurium*, *S. aureus*, and *C. difficile*, respectively) (Fig. 4a). Since the other species could compete with *B. subtilis*, we introduced antibiotics (ABX) to inhibit the growth of the target cells and enhance their eDNA release. The DNA sensor strains harbored antibiotic resistance genes and were thus resistant to the antibiotics. In the presence of 100 μg/mL spectinomycin, the DNA sensors displayed robust detection of *E. coli*, *S. typhimurium*, and *S. aureus* (Fig. 4b). The addition of spectinomycin was not required for *C. difficile* detection since the growth of *C. difficile* was negatively impacted by the presence of oxygen[27]. To confirm the transformation was mediated by the eDNA released from the target strain, DNase I (1 unit/mL) was added into the co-culture. The number of transformed cells was substantially lower in the presence of DNase I, indicating that

DNA detection occurred via natural transformation in the co-cultures (Fig. 4b). Antibiotic resistance is prevalent in microbiomes and may not be used universally as a treatment for the target cells. Therefore, we tested if heat treatment could be used to inhibit the growth of the target strain and enhance eDNA release. Incubation of *E. coli* at 90 °C for 10 min substantially enhanced the EC sensor detection limit in the co-culture ($5 \times 10^6$ cells per mL) compared to spectinomycin treatment (Fig. 4c). Therefore, heat treatment enabled efficient species detection without DNA purification.

To evaluate the robustness of the DNA sensing function for complex samples, we characterized the performance of the DNA sensors for multiplexed detection of spike-in bacteria in the presence of cecal contents derived from germ-free mice that were orally gavaged with a defined bacterial consortium (Methods). Cecal contents derived from mice contain diverse bacterial species, host cells, viruses and other chemical compounds (e.g. dietary factors), and thus can be used to evaluate the robustness of the DNA sensor to these factors. To this end, we introduced varying amounts of *E. coli* and *S. typhimurium* into 10 mg of mouse ceca. Samples were incubated at 90 °C for 10 min and transferred into a mixed culture of the EC-G and ST-R sensors (Methods; Fig. 4d). Our results demonstrated that both sensors detected spike-in *E. coli* and *S. typhimurium* cells in 10 mg mouse ceca without DNA extraction. In particular, the EC and ST sensors displayed a detection limit of 10⁷ cells per mL (10⁹ cells per gram) (Fig. 4d). In samples containing a single target species, high density of *E. coli* or *S. typhimurium* (10⁸ cells per mL) yielded higher

false positives for the multiplexed DNA detection (Supplementary Fig. 10). This suggests that further optimization of the DNA sensors may be needed in complex samples containing high target cell densities. In sum, these data demonstrate that the DNA sensor can perform multiplexed detection of target cells in a complex sample containing other biological factors (e.g. host cells and other bacteria).

## Discussion

Here we engineered the naturally competent bacterium *B. subtilis* to sense and respond to specific DNA sequences. DNA sensing can be achieved for purified DNA or eDNA released from pre-treated cells. We demonstrate that DNA sensing is specific and multiplexed sensing can be achieved in complex samples. Detection of species using a living cell-based DNA sensor strain enables versatile and programmable sensing of bacteria that does not rely on small molecule chemical or physical signals[28,29]. Since our circuit design is modular, customized sensors could be constructed in the future for the detection of sequences derived from diverse organisms including viruses, fungi and mammalian cells.

The living DNA sensors have potential for a wide range of DNA detection applications. The cell-based DNA detection is relatively simple and cost-effective. In comparison, other DNA detection methods such as next-generation sequencing and quantitative polymerase chain reaction require specialized instruments. In addition, *B. subtilis* spores are robust for potential long-term storage[30]. However, to achieve environmentally relevant concentrations, the sensitivity of the DNA sensor needs to be substantially enhanced. The sensitivity of the DNA sensor is determined by the transformation efficiency and background mutation rate of the negative growth selection module of the circuit. The DNA sensor sensitivity could be improved by modifications to the genetic circuit design[31,32], engineering of the strain[33], directed evolution[17], or selection of an alternative chassis with a higher transformation efficiency[34]. In *B. subtilis*, rational strain engineering[33] and directed evolution[17] could be performed to enhance transformation efficiency. To reduce the background mutation rate, the circuit design could be modified to include more efficient counter-selectable markers[31,32]. For example, if the background mutation rate was reduced from $10^{-6}$ (Fig. 2a) to $10^{-8}$ using a more efficient counter-selectable marker[31,32], pathogens with lower abundance such as *C. difficile* could be detected in fecal samples ($10^7$ cells per gram)[35]. Nevertheless, our current detection limit is close to the sensitivity range required for DNA detection during certain pathogen infection processes. For example, certain pathogens such as *Salmonella* in contaminated yolk ($10^{10}$ cells per gram)[36] or in ceca of infected chicken ($10^9$ cells per gram)[37] can achieve high densities close to the detection range of the DNA sensor ($10^9$ cells per gram for *Salmonella*, Fig. 4d). In addition, some naturally competent bacteria such as *Streptococcus pneumoniae* can achieve $10^{-1}$ transformation efficiency[34], which is substantially higher than our constructed DNA sensors ($10^{-5}$–$10^{-4}$, Fig. 2a). Using a suitable chassis with a high transformation efficiency, GFP expression could be observed at an earlier time, enabling a faster detection time[34].

We focused on the detection of diverse bacteria (i.e. different species) as opposed to characterizing DNA sensing performance of strain-level differences. Based on our results, we predict that the DNA sensors could differentiate target strains harboring conserved sequence with less than 77% nucleotide sequence similarity based on the stringent requirements of homologous recombination in *B. subtilis* (Fig. 2i–l). To achieve higher strain-level specificity, the landing pads could be composed of unique sequences with minimal conservation to other strains. For example, to differentiate *E. coli* MG1655 and *E. coli* DH10B, the landing pads could be designed using the *lac* operon or *leuLABCD* operon in MG1655, which is not present in DH10B (Δ*lacX74*, Δ*(ara leu) 7697*)[38]. To even achieve single nucleotide-level specificity, CRISPR could be incorporated into the genetic circuit design[39].

The power of a living cell-based sensor is illustrated by early detection of pathogens in situ prior to infection. *B. subtilis* has been shown to colonize or reside temporarily in diverse environments including soil and the mammalian gastrointestinal tract[40,41], enabling in situ DNA monitoring. For example, living DNA sensors could be introduced into the gastrointestinal tract or associated with plants to monitor microbiome dynamics by sensing and recording in real time[42]. The sensing mechanisms could also be coupled to the release of antimicrobials to target specific pathogens[43]. A recent study demonstrated that the naturally competent bacterium *Acinetobacter baylyi* can be engineered to detect tumor DNA in the mouse colon[39], demonstrating an additional potential application of in situ DNA detection. Since the cell-based DNA sensor can identify long homologous DNA sequences, the DNA sensor could also be repurposed to mine biosynthetic gene clusters from environmental DNA by targeting genes in particular pathways[44]. In sum, engineering DNA-sensing bacteria could open new avenues for both in vitro and in situ applications in the future.

## Methods
### Plasmid and strain construction
All DNA sensor strains were derived from *B. subtilis* PY79. Plasmids used or constructed in this work are listed in Supplementary Table 1. The pAX01-comK plasmid was purchased from the Bacillus Genetic Stock Center (BGSC ID: ECE222) to introduce $P_{xylA}$-*comK* to the *lacA* locus in *B. subtilis* PY79 by the MLS selection (1 μg/mL erythromycin from Sigma-Aldrich and 25 μg/mL lincomycin from Thermo Fisher Scientific), which can enhance the transformation efficiency in LB in the presence of xylose[17,45]. Fluorescent protein genes GFP(Sp), mCherry, and mTagBFP were cloned from plasmids pDR111_GFP(Sp)[46] (BGSC ID: ECE278), mCherry_Bsu[14] (BGSC ID: ECE756), and mTagBFP_Bsu[14] (BGSC ID: ECE745) to construct fluorescent reporter plasmids pOSV00170, pOSV00455 and pOSV00456, respectively. The fluorescent reporter was introduced into *ycgO* locus by selection in the presence of 5 μg/mL chloramphenicol (MilliporeSigma) and expressed from the IPTG-inducible promoter $P_{hyperspank}$. The kill switch plasmid pOSV00157 was composed of the repressor *lacI* and an IPTG-inducible toxin-antitoxin system $P_{hyperspank}$-*txpA-ratA* and can be introduced into the *amyE* locus by selection in the presence of 100 μg/mL spectinomycin (Dot Scientific). The toxin-antitoxin system *txpA-ratA* was amplified from *B. subtilis* 168 gDNA by polymerase chain reaction (PCR) and cloned onto plasmid pOSV00157.

*B. subtilis*, *E. coli*, *S. typhimurium*, *S. aureus* and *S. epidermidis* were all cultured at 37 °C in Lennox LB medium (MilliporeSigma). *C. difficile* and human gut species *A. caccae*, *B. thetaiotaomicron*, *C. asparagiforme*, *C. hiranonis* and *B. longum* were cultured at 37 °C in YBHI medium in an anaerobic chamber (Coy Laboratory). YBHI medium is Brain-Heart Infusion Medium (Acumedia Lab) supplemented with 0.5% Bacto Yeast Extract (Thermo Fisher Scientific), 1 mg/mL D-Cellobiose (MilliporeSigma), 1 mg/mL D-maltose (MilliporeSigma), and 0.5 mg/mL L-cysteine (MilliporeSigma). The gDNA of each species was extracted using DNeasy Blood & Tissue Kit (Qiagen). For *S. aureus* gDNA extraction, 0.1 mg/mL Lysostaphin (MilliporeSigma) was added in the pre-treatment step in combination with enzymatic lysis buffer (Qiagen). Bacterial strains are listed in Supplementary Table 2.

The target sequences *xdhABC* were PCR amplified from *E. coli* MG1655 gDNA (NCBI Reference Sequence: NC_000913.3, location: 3001505-3004004 and 3004005-3006504), *sipBCDA* from *Salmonella enterica* serovar Typhimurium LT2 ATCC 700720 (NCBI Reference Sequence: NC_003197.2, Location: 3025979-3028478 and 3028479-3030978), *hemEH* from *S. aureus* DSM 2569 (GenBank: LHUS02000002.1, Location: 553-2770 and 2864-5638), and *pheST* from *C. difficile* DSM 27147 (GenBank: FN545816.1, Location: 770923-773144 and 773157-775686) to construct a set of plasmids (pOSV00169, pOSV00205, pOSV00206, pOSV00207, pOS00208,

pOSV00292, pOSV00459 and pOSV00475) using restriction enzymes BamHI-HF (New England Biolabs) and EcoRI-HF (New England Biolabs) or Golden Gate Assembly Mix (New England Biolabs). These plasmids were derived from the kill switch plasmid pOSV00157 where the target sequences were introduced to the upstream and downstream of repressor *lacI* and toxin-antitoxin system $P_{hyperspank}$-*txpA-ratA*. DNA sequences of genetic parts are listed in Supplementary Table 3.

All plasmids were constructed using *E. coli* DH5α and transformed into *B. subtilis* using MC medium[47]. MC medium is composed of 10.7 g/L potassium phosphate dibasic (Chem-Impex International), 5.2 g/L potassium phosphate monobasic (MilliporeSigma), 20 g/L glucose (MilliporeSigma), 0.88 g/L sodium citrate dihydrate (MilliporeSigma), 0.022 g/L ferric ammonium citrate (MilliporeSigma), 1 g/L Oxoid casein hydrolysate (Thermo Fisher Scientific), 2.2 g/L potassium L-glutamate (MilliporeSigma), and 20 mM magnesium sulfate (MilliporeSigma). Plasmid DNA was extracted from *E. coli* DH5α using Plasmid Miniprep Kit (Qiagen). *B. subtilis* was inoculated into MC medium and incubated at 37 °C for 4 h. Extracted plasmids were added into the *B. subtilis* culture and incubated at 37 °C for another 2 h before plating. Transformed *B. subtilis* were selected on LB agar plates with selective antibiotics (spectinomycin, MLS and kanamycin). Double crossover was verified for single colonies by the replacement of a different antibiotic resistance gene at the integration locus (*lacA::specR*, *ycgO::erm* or *amyE::kan*).

### DNA detection using cell-based sensors

DNA sensor strain was inoculated from the −80 °C glycerol stock into LB medium and incubated at 37 °C with shaking (250 rpm) for 14 h. On the next day, the OD600 of overnight culture was measured by NanoDrop One (Thermo Fisher Scientific) and diluted to OD600 0.1 in 1 mL LB in a 14 mL Falcon™ Round-Bottom Tube (Thermo Fisher Scientific) supplemented with 50 mM xylose (Thermo Fisher Scientific) and 100 μg/mL spectinomycin (Dot Scientific). gDNA was quantified by the Quant-iT dsDNA Assay Kit (Thermo Fisher Scientific) and introduced into the DNA sensor culture. The DNA sensor culture was incubated at 37 °C with shaking (250 rpm) for 10 h for transformation. Transformed sensors (5 μL) was plated onto a 12-well plate (Thermo Fisher Scientific) for selection. Each well contained 1 mL LB agar supplemented with 2 mM IPTG (Bioline), 5 μg/mL chloramphenicol (MilliporeSigma), and MLS (1 μg/mL erythromycin from Sigma-Aldrich and 25 μg/mL lincomycin from Thermo Fisher Scientific). GFP-expressing colonies were imaged using an Azure Imaging System 300 (Azure Biosystems) by Epi Blue LED Light Imaging with 50 millisecond exposure time. Colony forming unit (CFU) was counted manually. Transformation efficiency is defined as the ratio of CFU on selective plates (transformed *B. subtilis* with GFP expression) to the CFU on non-selective plate (total *B. subtilis*). To count CFU of total *B. subtilis*, cell culture was serially diluted in phosphate-buffered saline (PBS) (Dot Scientific) and plated onto LB agar plates containing 5 μg/mL chloramphenicol and MLS. Since the total *B. subtilis* CFU was similar with or without DNA supplement (Supplementary Fig. 7b), CFU of transformed cells can be simply used to indicate transformation efficiency of individual sensor strains. To confirm that the DNA detection was via the synthetic genetic circuit, gDNA of fluorescent transformed colonies was extracted by DNeasy Blood & Tissue Kit (Qiagen). Primers targeting the landing pads of *E. coli* sensor (EC_FW and EC_RV) were used to PCR amplify and sequence the landing pads from the gDNA of the transformed *E. coli* sensor to confirm the loss of the repressor *lacI* and toxin-antitoxin system *txpA-ratA*. The sequence of the PCR product was confirmed by Sanger sequencing. Primers targeting the repressor *lacI* and toxin-antitoxin system *txpA-ratA* (lacI_txpA_FW and lacI_txpA_RV) were used to confirm the mutations in the escape mutants. Primer sequences are listed in Supplementary Table 4.

To determine the sensitivity of sensors, serially diluted DNA from 1500 ng/mL to 1 ng/mL was supplemented to the sensor culture

for 10-h transformation, and transformed cell culture was transferred into liquid LB medium with a 1:20 dilution ratio in a 96-well black and clear-bottom CELLSTAR® microplate (Greiner Bio-One). To test the specificity towards gDNA from different strains, 100 ng/mL purified DNA was used for transformation. Liquid LB medium was supplemented with 2 mM IPTG (Bioline), 5 μg/mL chloramphenicol (MilliporeSigma), and MLS. Plate was sealed with Breathe-Easy Adhesive Microplate Seals (Thermo Fisher Scientific) and incubated in the SPARK Multimode Microplate Reader (TECAN) at 37 °C with shaking for time-series OD600 and GFP fluorescence measurements. A threshold of GFP fluorescence 400 was used to determine the detection time for different DNA or different concentrations. The threshold was selected in the linear region of the fluorescence curve where the difference in the detection time in response to different gDNA concentrations or absence of gDNA was not substantially affected by the choice of threshold. An unpaired *t*-test was used to determine if the detection time of specific DNA concentration was different from the condition without DNA with 4 technical replicates. The detection limit is the lowest DNA concentration with a detection time statistically different from the background mutations. A linear function was fit to the log transformed gDNA concentration versus mean detection time to determine significant differences. The DNA mass per mL in detection limit (1 ng, 62.5 ng, 4 ng, and 16 ng) and genome size (4639675 bp, 4857450 bp, 2827820 bp, and 4153430 bp) were converted to the chromosome copy number by NEBioCalculator for *E. coli* ($2.10 \times 10^5$), *S. typhimurium* ($1.25 \times 10^7$), *S. aureus* ($1.38 \times 10^6$), and *C. difficile* ($3.75 \times 10^6$), respectively. Matlab R2019a (9.6.0.1072779) were used to generate plots of OD600 and fluorescence and perform unpaired *t*-tests.

### Bioinformatic analysis of target DNA sequences

To analyze if the target sequence is conserved for different strains in the same species, we searched the 5000 bp of *E. coli xdhABC*, *S. typhimurium sipBCDA*, *S. aureus hemEH*, and *C. difficile pheST* DNA sequence (Supplemental Table 3) in the NCBI Nucleotide Collection Database. Nucleotide BLAST was optimized for somewhat similar sequences (blastn). The search was specified to taxid 561 for *E. coli*, taxid 28901 for *S. enterica*, taxid 1280 for *S. aureus*, and taxid 1496 for *C. difficile*. The percentage of strains with conserved sequences (more than 95% identity similarity and 95% coverage) is 99% ($N = 3462$), 94.3% ($N = 2078$), 95.6% ($N = 1488$), and 96.4% ($N = 139$) for *E. coli*, *S. typhimurium*, *S. aureus*, and *C. difficile*, respectively. To analyze if the target sequence is conserved in other species, the same search was performed excluding the same species. Homologs with varying identity similarity and coverage were found in different species for the target sequences *E. coli xdhABC* ($N = 5000$), *S. typhimurium sipBCDA* ($N = 117$), *S. aureus hemEH* ($N = 2993$), and *C. difficile pheST* ($N = 5000$). Scatter plots of homology coverage and identity similarity for each strain other than the target species was displayed (Fig. 2i, k; Supplemental Fig. 3). Accession date of data is 2022-08-19.

### Multiplexed DNA detection using cell-based sensors

Overnight cultures of DNA sensor strains EC-G, ST-R and SA-B were diluted to OD600 0.1, 0.1, and 0.01, respectively, in one single culture containing 1 mL LB supplemented with 50 mM xylose (Thermo Fisher Scientific) and 100 μg/mL spectinomycin (Dot Scientific). Different combinations of gDNA extracted from *E. coli*, *S. typhimurium*, and *S. aureus* (200 ng/mL each) were added into the mixed cultures containing the three DNA sensor strains in 14 mL Falcon tubes (Thermo Fisher Scientific). The DNA sensor strains were incubated at 37 °C with shaking (250 rpm) for 10 h and 5 μL of transformed cell culture was plated onto 12-well plates (Thermo Fisher Scientific). In these plates, each well contained 1 mL LB agar supplemented with 2 mM IPTG, 5 μg/mL chloramphenicol, and MLS. Plates were

incubated at 37 °C overnight. On the next day, individual wells of 12-well plate were imaged using Nikon Eclipse Ti-E Microscope. Brightfield images were collected at 4X magnification using the built-in transilluminator of the microscope. Fluorescence images of colonies were collected with the epifluorescence light source X-Cite 120 (Excelitas) and standard band filter cubes including Texas Red (Excitation: 560/40 nm, Emission: 630/70 nm, Nikon), GFP (Excitation: 470/40 nm, Emission: 525/50 nm, Nikon), and BFP (Excitation: 395/25 nm, Emission: 460/50 nm, Chroma) to image RFP, GFP, and BFP, respectively. Images were processed with $8 \times 8$ binning for each image. The exposure times for RFP, GFP, and BFP were 7 ms, 1.5 ms, and 1.5 ms, respectively. Complete images of each well were generated from multipoint images after scanning a $24 \times 24$ mm area. Once the full images were assembled, each of the four channels were mapped to unity by the minimum and maximum pixel values of all images for that channel using ImageJ (v1.53c). Colonies of different colors were counted manually to determine transformation efficiency.

To assemble a multi-species microbial community for evaluating multiplexed detection, bacterial species *A. caccae*, *B. thetaiotaomicron*, *B. longum*, *C. asparagiforme*, *S. typhimurium*, and *S. aureus* were inoculated from the −80 °C glycerol stock into YBHI medium and incubated at 37 °C overnight in an anaerobic chamber (Coy Laboratory). On the next day, cell culture of each strain was diluted to OD600 0.01 into YBHI in one single culture (Passage 0) and incubated for 24 h (Passage 1). The overnight cell culture containing 6 species was diluted to OD600 0.1 into a fresh YBHI and incubated for another 24 h (Passage 2). At each passage, a cell pellet was collected for DNA extraction using a DNeasy Blood & Tissue Kit (Qiagen). To extract *S. aureus* gDNA, 0.1 mg/mL Lysostaphin (MilliporeSigma) was added in the pre-treatment step in combination with enzymatic lysis buffer (Qiagen). Purified DNA was stored at −20 °C for *B. subtilis* transformation or NGS analysis.

To characterize microbial community compositions by NGS, the V3-V4 region of the 16 S rRNA gene was PCR amplified from extracted DNA using custom dual-indexed primers on a 96-well PCR plate (detailed method described in Clark et al. Nat. Comm., 2021[48]). PCR products from each well were pooled and purified using a DNA Clean & Concentrator kit (Zymo). The resulting library was sequenced on an Illumina MiSeq using a MiSeq Reagent Kit v2 (500-cycle) to generate $2 \times 250$ paired-end reads. Sequencing data were demultiplexed using Basespace Sequencing Hub's FastQ Generation program. Sequences of each strain were mapped to the 16 S rRNA reference database using custom Python scripts. The reference database of the V3–V4 16 S rRNA gene sequences was created using consensus sequences from Sanger sequencing data of monospecies cultures. Relative abundance was calculated as the read count mapped to each species divided by the total number of reads of all species.

To detect *S. typhimurium* and *S. aureus* gDNA in the community DNA by cell-based DNA sensors, extracted community DNA (1000 ng/mL) was added to a mixed culture of SA-G (OD600 0.1) and ST-R sensors (OD600 0.1). The cell culture was incubated at 37 °C with shaking (250 rpm) for 10 h and 5 μL of cell culture was plated onto 12-well plates (Thermo Fisher Scientific) containing 1 mL LB agar supplemented with 2 mM IPTG, 5 μg/mL chloramphenicol, and MLS. Plates were incubated at 37 °C overnight. On the next day, each well was imaged using a Nikon Eclipse Ti-E Microscope. Brightfield and fluorescence images of GFP and RFP were collected at 4X magnification using the same procedure described above for multiplexed detection. Colony numbers of different colors were counted manually. The average colony number of 3 technical replicates at Day 1 and Day 2 was divided by the average colony number at Day 0 to calculate the fold change of *S. typhimurium* and *S. aureus* abundance over time, which was compared to the 16S rRNA gene next-generation sequencing results.

## Direct detection of target species using cell-based sensors

The target strains *E. coli*, *S. typhimurium*, and *S. aureus* and DNA sensor strains were inoculated from the −80 °C glycerol stock into LB medium and incubated at 37 °C with shaking (250 rpm) for 14 h. *C. difficile* was inoculated in YBHI medium and incubated in an anaerobic chamber (Coy Laboratory). On the next day, cell cultures of sensor and target strain were diluted (OD600 0.1 each) into a single culture containing 1 mL LB supplemented with 50 mM xylose with or without 100 μg/mL spectinomycin in 14 mL Falcon tubes (Thermo Fisher Scientific). To determine the initial CFU of target bacteria in the mixed culture, overnight culture of target strains was diluted in PBS (Dot Scientific) and plated onto LB agar plates or YBHI agar plates. The target strain abundance was $1.22 \times 10^8$ CFU/mL, $1.07 \times 10^8$ CFU/mL, $3.2 \times 10^8$ CFU/mL, and $1.1 \times 10^7$ CFU/mL for *E. coli*, *S. typhimurium*, *S. aureus*, and *C. difficile*, respectively. The sensor and target strains were co-cultured at 37 °C with shaking (250 rpm) for 10 h, and 5 μL of cell culture was plated onto 12-well plates (Thermo Fisher Scientific) containing 1 mL LB agar, 2 mM IPTG, 5 μg/mL chloramphenicol, and MLS. The 12-well plates were incubated overnight at 37 °C. On the next day, fluorescent colonies were imaged using an Azure Imaging System c300 (Azure Biosystems) by the Epi Blue LED Light Imaging with 50 millisecond exposure time. Colonies with GFP fluorescence expression were counted manually. To determine if the appearance of colonies was due to natural transformation, 1 μL (1 unit) DNase I (Thermo Fisher Scientific) was added to the 1 mL co-culture, which can completely degrade 1 μg of plasmid DNA in 10 min at 37 °C according to the manufacturer's specification.

To improve the detection efficiency, overnight culture of *E. coli* was incubated at 90 °C in digital dry baths/block heaters (Thermo Fisher Scientific) for 10 min, placed on ice for 3 min, and diluted to the sensor culture containing 1 mL LB, 50 mM xylose, and sensor strain (OD600 0.1) without spectinomycin for detection. To perform multiplexed detection for spike-in *E. coli* and *S. typhimurium* in mice ceca, 10 mg ceca was resuspended in 100 μL LB supplemented with 50 mM xylose in 1.7 mL Eppendorf tubes (Dot Scientific). A low amount of mouse cecal contents (10 mg per mL) was used to minimize a negative effect of the cecal contents on *B. subtilis* growth and transformation efficiency. Varying amounts of *E. coli* and *S. typhimurium* were spiked into resuspended liquid ceca content. The samples were incubated at 90 °C for 10 min, placed on ice for 3 min, and transferred to the mixed culture of EC-G and ST-R sensors (1 mL LB, 50 mM xylose, and OD0.1 for each sensor strain) for multiplexed detection. Cecal contents were collected from germ-free mice following protocols approved by the University of Wisconsin-Madison Animal Care and Use Committee. Briefly, 8-week-old C57BL/6 gnotobiotic male mice (wild-type) were orally gavaged with 8 human gut bacteria - *Dorea formicigenerans*, *Coprococcus comes*, *Anaerostipes caccae*, *Bifidobacterium longum*, *Bifidobacterium adolescentis*, *Bacteroides vulgatus*, *Bacteroides caccae*, and *Bacteroides thetaiotaomicron* via oral gavage. After 4 weeks of colonization, mice were euthanized for cecal collection. The mouse cecal contents were stored at −80 °C prior to the introduction of spike-in bacteria.

## Statistics and reproducibility

A sample size of 3 was chosen for all characterizations of the cell-based DNA sensors in biological replicates (three independent transformations) or technical replicates (one transformation with three separate colony forming unit measurements on the same sample). Transformation results were highly similar across either 3 biological replicates or 3 technical replicates. A sample size of 4 was chosen for plate reader (absorbance at 600 nm and fluorescence measurements) measurements specifically due to the observed variability across technical replicates. An unpaired, two-tailed Student *t*-test was performed using Matlab R2019a (9.6.0.1072779) to calculate *p*-values to determine if the detection time in the presence of varying concentrations of

DNA was significantly different from the detection time in the absence of DNA based on the technical replicates ($N = 4$). Using a sample size of 4, we observed a linear relationship between the observed detection times of cell-based DNA sensors and the log transformed gDNA concentrations (Fig. 2d–g).

## Reporting summary

Further information on research design is available in the Nature Portfolio Reporting Summary linked to this article.

## Data availability

Source data for Figs. 1–4 are provided and can also be downloaded from the BioStudies database (https://www.ebi.ac.uk/biostudies/studies/S-BSST1044). Source data are provided with this paper.

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

## Acknowledgements
We would like to thank Dr. Scott Coyle and Zhejing Xu for the assistance of microscopic imaging. We also thanks Yiyi Liu in the Venturelli lab for providing the mice ceca with defined microbial communties. This work was supported by the Defense Advanced Research Projects Agency (DARPA) Grant HR0011-19-2-0002 and National Institutes of Health Biomedical Imaging and Bioengineering R01EB030340.

## Author contributions
Y.-Y.C., B.M.B. and O.S.V. conceived the research. Y.-Y.C., and Z.C. performed the experiments. Y.-Y.C., Z.C., and X.C. constructed the sensors. T.D.R. processed the microscopic images. T.G.F. assisted with the strain construction. Y.-Y.C. and O.S.V. wrote the manuscript. O.S.V. secured the funding.

## Competing interests
O.S.V., Y.-Y.C., and Z.C. have filed a nonprovisional application with the US Patent and Trademark Office on the cell-based DNA detection method (U.S. Nonprovisional Patent No. 18/067,194). The other authors declare that they have no competing interests.
