## [Peer Review File · Nature Communications]

Reviewers' Comments:

Reviewer #3:

Remarks to the Author:

I feel the authors have adequately addressed my comments from the first (and technically second) round of review. The focus of the manuscript has changed towards detection of bacteria without need for extraction of DNA. This minimises the need for practicality and optimisation of the method. It also minimises the need to discuss alternative assay formats (qPCR, CRISPR, etc), although it might be useful and interesting to do so and it was mentioned by multiple reviewers.

I also note that some of the other reviewers comment that the spiked cecal samples might not go far enough to demonstrate an application with a real world sample. This is something for the authors to consider.

Overall, this is a unique idea and I think it is a contribution to the literature. It may not yet be practical for deployment, but it has potential with future work to become so.

Reviewer #4 (Remarks to the Author):

I feel the authors have adequately addressed my comments from the first (and technically second) round of review. The focus of the manuscript has changed towards detection of bacteria without need for extraction of DNA. This minimises the need for practicality and optimisation of the method. It also minimises the need to discuss alternative assay formats (qPCR, CRISPR, etc), although it might be useful and interesting to do so and it was mentioned by multiple reviewers.

We thank the reviewer for the comment. We discussed the potential *in vitro* applications, current limitation, and future modifications in the Discussion (highlighted text).

I also note that some of the other reviewers comment that the spiked cecal samples might not go far enough to demonstrate an application with a real world sample. This is something for the authors to consider.

We thank the reviewer for this comment. We discuss the potential for in situ applications in the future in our Discussion section (highlighted text) and also describe how the DNA sensor sensitivity needs to be improved for these applications by reducing the background mutation frequency.

Overall, this is a unique idea and I think it is a contribution to the literature. It may not yet be practical for deployment, but it has potential with future work to become so.

We thank the reviewer for his/her appreciation of our work. Our future plan is to improve the performance of the cell-based DNA sensor and apply this technology for real-world applications.